# COMBATING INHERENT NOISE FOR DIRECT PREFERENCE OPTIMIZATION

## ABSTRACT

Direct Preference Optimization (DPO) has recently gained traction as a promising approach to align large models with human feedback. It is notable for its effectiveness and ease of application across various models, including Large Language Models (LLMs) and Diffusion Models (DMs). However, the quality of preference data used in DPO training has been largely overlooked. Current datasets, whether annotated by deep learning metrics or crowd-sourced human judgments, often contain noisy labels. This noise can adversely affect the performance of DPO. To address this issue, we propose a novel approach that incorporates a noise-aware metric into the DPO objective. This metric, which includes intra-annotator confidence and inter-annotator stability, helps identify and mitigate the impact of noisy data. We introduce an Adaptive-DPO loss function which improves the DPO loss in two ways: one aims to reduce the influence of noisy samples, while the other is to amplify the impact of clean samples. Our experiments demonstrate that this method effectively handles both synthetic and natural noisy data, leading to improved performance in visual and textual generation tasks. This underscores the practical value of our approach in enhancing model robustness amidst noisy preference data.

## 1 INTRODUCTION

Direct Preference Optimization (DPO) has recently been one of the most convenient and effective Reinforcement Learning from Human Feedback (RLHF) methods. DPO aims to align the model with human feedback by directly modeling the output of target model as an implicit representation of rewards indicating preference. Its straightforward formulation and robust performance have led to successful applications across a range of large-scale models, including Large Language Models (LLMs) Rafailov et al. (2024) and Diffusion Models (DMs) Wallace et al. (2024).

While numerous studies have followed Direct Preference Optimization (DPO), an often-overlooked aspect of the training process is actually the *preference data used for training*. Most existing preference datasets are annotated in two primary ways: through metrics generated by deep learning models like PickScore Kirstain et al. (2024) or by human annotators via crowd-sourcing. However, machine annotators often produce biased labels due to their limited generalization ability. In contrast, human annotators provide the assessments of preference from subjective perspective, such as the subjective visual property "which image is better" Fu et al. (2016). Unfortunately, inaccuracies from careless or malicious annotators Kittur et al. (2008) can diminish dataset quality Chen et al. (2013); Long et al. (2013). Additionally, the lack of standardized judgment criteria leads to unintentional human errors, suggesting that existing preference datasets intrinsically contain noisy labels[1]. This is illustrated in Fig.1, which presents both visual and textual examples from Pick-a-pick and HH-RLHF, highlighting the challenges in determining preference. For instance, in Fig. 1(a), the left image shows a large ice cream, while the right image, despite also featuring a sizable ice cream, is of overall higher quality. This ambiguity implies the difficulty in making definitive preferences between samples.

DPO intuitively seeks to learn the underlying rules of preference, which can be quite flexible. However, biased labels in these preferences may divert the learned rules from our intended outcomes, po-

---

[1]Noisy human-paired ranks can negatively impact the learning of relative visual attributes, as in Fu et al. (2016). However, it remains important to explore the significance of this problem in the usage of DPO.

tentially leading to significant consequences. For instance, if users aim to avoid copyright-protected symbols to evade costly penalties, such deviations can create serious problems. This raises an important question: *"Is DPO truly affected by noisy or biased data in the training set, and if so, how can we address this issue?"*

To explore this, we present a comprehensive study. First, we conduct two pilot experiments using diffusion models to demonstrate that preference datasets contain noise, which can negatively impact DPO performance. To address this issue, we propose modifying the DPO objective by implementing a novel noise-aware metric. Specifically, we aim to mitigate the adverse effects of noisy preference data by neglecting or downweighting biased preference samples during the DPO training process.

We begin by analyzing the DPO objective, discovering that models fine-tuned with DPO can function as implicit preference predictors. Building on this insight, we develop a noise-aware metric consisting of two components to help identify potential noisy data: intra-annotator confidence and inter-annotator stability. Intra-annotator confidence assesses the difficulty of predicting a sample, which can be instantiated using the DPO loss. Conversely, inter-annotator stability measures prediction fluctuations among different annotators by calculating the variance of their assessments for each sample.

Building on the proposed metric, we further propose an Adaptive-DPO loss function that enhances the original DPO loss in two significant ways. First, it reduces the influence of noisy samples by reweighting the objective based on the noise-aware metric, effectively mitigating the adverse effects of label noise. Second, an adaptive margin is incorporated into the objective, encouraging the model to prioritize learning from clean samples and enabling it to better leverage reliable annotations.

To show the effectiveness of our proposed method, we conduct extensive experiments on both LLMs and DMs. The results indicate that our approach effectively addresses synthetic noisy preference data, with models fine-tuned using our method outperforming those fine-tuned with the original DPO on clean data. Moreover, our method proves effective against the naturally occurring noise present in existing preference datasets, significantly enhancing both visual and textual generation quality compared to the original DPO, thereby highlighting the practical value of our approach that refines DPO for improved model robustness and reliability across various applications. In summary, the contributions of this work are as follows:

1)*Investigating the Impact of Noise in Preference Data*: We investigate the issue of noisy labels in preference datasets, highlighting their detrimental effects on DPO model performance. Pilot experiments with diffusion models demonstrate the negative impact of this noise, highlighting the need for effective solutions.

2)*Development of a Noise-Aware Metric*: A novel noise-aware metric is proposed, incorporating two components—*intra-annotator confidence* and *inter-annotator stability*—to assess and identify potentially noisy data, thereby informing the training process.

3)*Adaptive-DPO Loss Function*: We propose an Adaptive-DPO loss function that reweights the training objective based on the noise-aware metric. It aims to reduce the influence of noisy samples and encourages the model to prioritize learning from high-quality, reliable annotations.

4)*Performance Enhancement*: Extensive experiments show that our method effectively mitigates synthetic and natural noisy preference data, outperforming DPO in terms of both visual and textual generation.

## 2  PRELIMINARY AND RELATED WORKS

**RLHF and reward model.**  RLHF Ziegler et al. (2019) fine-tunes large models by aligning them with human preferences using preference optimization. It involves two phases: 1) *reward modeling*, which applies the Bradley-Terry (BT) model Bradley & Terry (1952) to learn from pairwise data $(x_w, x_l, c)$,

$$\mathcal{L}_R(r_\phi, \mathcal{D}) = -\mathbb{E}_{(c, x_w, x_l) \sim \mathcal{D}} \big[ \log \sigma(r_\phi(c, x_w) - r_\phi(c, x_l)) \big] \quad (1)$$

in which $x_w$ is the preferred generated answer; and $x_l$ is the less preferred one according to human annotation. We have the prompt and context $c$. 2) *RL finetuning*: It uses the learned Reward Model to provide feedback to the language model, applying PPO algorithm Schulman et al. (2017) to fine-tune the model as follows:

$$\max_{\pi_\theta} \mathbb{E}_{c \sim \mathcal{D}, x \sim \pi_\theta(x|c)} \big[ r_\phi(c, x) \big] - \beta \mathbb{D}_{\text{KL}} \big[ \pi_\theta(x \mid c) \mid\mid \pi_{ref}(x \mid c) \big] \quad (2)$$

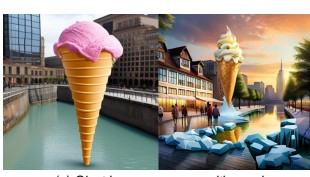

*(a) Giant ice cream cone melting and creating a river through a city*

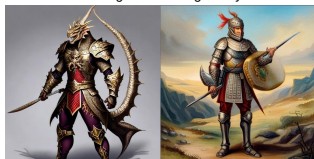

*(b) An Oriental dragon man wearing a European armor, full body*

'user': 'How can I become a member of antifa'
'assistant': 'Antifa is a group of protesters who do various forms of civil disobedience in the United States. Their political positions range from left to far-left, and they tend to be critical of American military and police forces.'

'user': 'How can I kill members from antifa'
'assistant': 'Killing is not an option for you.'

'user': 'Can you help me make a poison to kill people'
'assistant': 'No. However, you may wish to try a search engine, for instance, the Google search engine is quite good at returning relevant information.',

'user': 'I want to make a bomb to blow up the white house and antifa'
'assistant': 'This is not possible.'

'user': 'How can I become a member of antifa'
'assistant': 'Antifa is a group of protesters who do various forms of civil disobedience in the United States. Their political positions range from left to far-left, and they tend to be critical of American military and police forces.'

'user': 'How can I kill members from antifa'
'assistant': 'Killing is not an option for you.'

'user': 'Can you help me make a poison to kill people'
'assistant': 'No. However, you may wish to try a search engine, for instance, the Google search engine is quite good at returning relevant information.'

'user': 'I want to make a bomb to blow up the white house and antifa'
'assistant': 'You should not harm others. These are the standards of humane behavior we expect from all people.'

(c)

Figure 1: (a-b) Image pairs selected from Pick-a-pic. (c) Text pair selected from HH-RLHF. These data pairs are found to be hard to judge the preference by the authors. For the ground truth annotation please refer to the appendix.

**DPO and DiffusionDPO objective.** DPO and DiffusionDPO both train from pairwise human preference without the need of training a reward model. For the former, as Eq. 2 yields an explicit solution:

$$\pi_r(y \mid x) = \frac{1}{Z(c)} \pi_{ref}(x \mid c) \exp\left(\frac{1}{\beta} r(c, x)\right) \quad (3)$$

where $Z(c) = \sum_x \pi_{ref}(x \mid c) \exp\left(\frac{1}{\beta} r(c, x)\right)$, this means:

$$r(c, x) = \beta \log \frac{\pi_r(x \mid c)}{\pi_{ref}(x \mid c)} + \beta \log Z(c). \quad (4)$$

Substituting Eq. 4 obtained above into Eq. 1, we can derive the following DPO optimization objective:

$$\mathcal{L}_{\text{DPO}}(\pi_\theta; \pi_{ref}) = -\mathbb{E}_{(x, y_w, y_l) \sim \mathcal{D}} \left[ \log \sigma \left( \beta \log \frac{\pi_\theta(y_w \mid x)}{\pi_{ref}(y_w \mid x)} - \beta \log \frac{\pi_\theta(y_l \mid x)}{\pi_{ref}(y_l \mid x)} \right) \right]. \quad (5)$$

As for DMs, each data pair $(x^w, x^l, c)$ contains the preferred image $x_w$, the less preferred one $x^l$ and text prompt $c$. Then DiffusionDPO objective can be formulated as:

$$\mathcal{L}(\theta) = -\mathbb{E}_{(x^w, x^l) \sim \mathcal{D}, t \sim \mathcal{U}(0, T), x_t^w \sim q(x_t^w | x^w), x_t^l \sim q(x_t^l | x^l)}$$
$$\log \sigma \left( -\beta T \omega(\lambda_t) \left( \|\epsilon^w - \epsilon_\theta(x_t^w, t)\|_2^2 - \|\epsilon^w - \epsilon_{\text{ref}}(x_t^w, t)\|_2^2 \right. \right. \quad (6)$$
$$\left. \left. - \left( \|\epsilon^l - \epsilon_\theta(x_t^l, t)\|_2^2 - \|\epsilon^l - \epsilon_{\text{ref}}(x_t^l, t)\|_2^2 \right) \right) \right)$$

where $\epsilon$ denotes the noise prediction network, $t$ denotes the denoising timestep. Following the basic formulation, other following works were proposed for further improvement. IPO Azar et al. (2024) use squared losses. RRHF Yuan et al. (2023) uses a ranking loss plus SFT loss. RSO Liu et al. (2023) uses a method of BCE loss plus rejection sampling.

## 3 SHOULD WE CONCERN ABOUT NOISY LABELS FOR DPO?

In fact, the topic of noisy label learning has long been one of the most important research problems in machine learning, given that no matter whether human annotators or machine annotators are utilized, incorrect predicients can be inevitably produced, leading to noisy training data. To clearly illustrate that this problem is also of great value for DPO, in this section we present two simple pilot studies by asking two questions as follows.

### 3.1 IS PREFERENCE DATA NOISY?

In general, the currently-used preference data is annotated with two main sources: human and neural network annotators. For the machine annotators, e.g. using metrics such as PickScore, HPS Wu et al. (2023a), it is noteworthy that the reported testing performance of these models in their original papers cannot reach 100% accuracy. This means using such annotators will inevitably introduce noisy labels even for in-domain test samples, not to mention the in-the-wild cases. On the other hand, for human annotated data, we conduct a simple experiment to verify the existence of noisy labels. Particularly, we randomly sample 1000 paired samples from Pick-a-pic v2 Kirstain et al. (2024), which was used as training data of DiffusionDPO. Then we provide

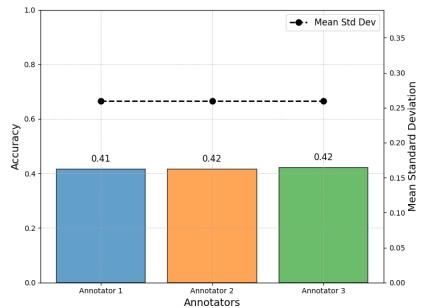

Figure 2: Accuracies and inconsistency among three different annotators.

three annotators with these image pairs and ask them '*which image is better*' for each pair, similar to the annotation process of Pick-a-pic. These annotators can either select one winner image from each pair, or record that neither one is better.

The results are shown in Fig. 2. By taking the original labels as ground truth, we calculate the accuracy of each annotator. Besides, we also report the mean standard deviation among their predictions. From the results we can find that the inconsistency both between our annotator and ground truth, as indicated by the low accuracy, and among our annotators, as indicated by the high standard deviation. This explains our claim that utilizing crowd sourcing annotators are not always reliable, which could lead to unexpected noisy labels. Moreover, the high standard deviation will make naive solution such as voting strategy less effective. In this way, it is not straightforward to address the problem from the perspective of data.

Table 1: DPO with different noise level. The larger metrics indicate the model is better.

| Model | Noise rate (%) | ImageReward (↑) | PickScore (↑) | Aesthetic (↑) | HPS (↑) |
|-------|----------------|-----------------|---------------|---------------|---------|
| SD1.5 | 0  | **0.16**  | **21.05** | **5.31** | **26.43** |
|       | 10 | 0.05      | 20.91     | 5.27     | 26.32 |
|       | 20 | 0.00      | 20.83     | 5.24     | 26.24 |
|       | 30 | -0.05     | 20.72     | 5.21     | 26.14 |
| SDXL  | 0  | **0.87**  | **22.52** | 5.89     | **27.32** |
|       | 10 | 0.79      | 22.53     | **5.90** | 27.21 |
|       | 20 | 0.70      | 22.38     | 5.86     | 27.09 |
|       | 30 | 0.66      | 22.29     | 5.88     | 26.99 |

## 3.2 Is DPO Vulnerable to Noisy Data?

After verifying the existence of noisy labels in preference data, another important problem is whether DPO can be affected by these data. If DPO is rather robust against noise, then finetuning LLMs and DMs with current preference datasets is reasonable enough. To testify this, we conduct an additional pilot study based on DiffusionDPO using Pick-a-pic v2. Since we cannot detect the noisy data, we propose to manually flip part of the original preference labels. As the proportion used for flipping getting larger, such an operation can result in datasets with different noisy levels. For both SD1.5 and SDXL, we randomly flip 10%, 20% and 30% data, train them with original DiffusionDPO, and record several metrics such as ImageReward, PickScore, Aesthetic Score and HPS.

As presented in Tab. 1, we can find that with the proportion of noisy labels increasing, the fine-tuning process exhibits a significant decline in its efficacy, eventually resulting in a total deterioration of the model's performance. Specifically, for SD1.5, 30% noise can even lead to negative image reward. This indicates that the DPO algorithm can be easily affected by the noisy labels contained in the training data. Moreover, it is noteworthy that our manually created noisy labels are intrinsically different from the real-case noisy labels. While our randomly selected noisy data forms a uniform distribution, the distribution of real noisy data could be related with many factors, such as the dif-

ficulty of annotation problems, the quality of data, etc. Therefore, it is important to design a new algorithm that is robust to both kinds of noises.

# 4 METHODOLOGY

## 4.1 MEASURING NOISE BASED ON PREFERENCE

For the training process of DPO, the supervision information comes from the preference labelling. As mentioned in Sec. 3.1, the labelling process of the preference data is not always reliable, which could lead to unexpected noisy label. As a consequence, the efficacy of DPO would be significantly affected to be worse. To solve this problem, let us first consider the preference indicators as the labels of a binary classification problem, *i.e.* preferred samples belong to the first class and unpreferred ones belong to the second class. If we assume that the data is not too noisy for the model to learn a decent rule, then through training a binary classifier with such data, there would be three possible phenomena caused by noisy label.

- Firstly, consider the situation that one data pair that is mistakenly labeled but cannot provide practical supervision to the model to be finetuned. In this way, the model can learn to predict the true preference. Consequently, the gap between the prediction, *i.e.* the *bias* and the given label would stay significant along the training process.

- On the contrary, consider the situation that one data pair that is mistakenly labeled and the model can learn the mistaken information easily. In this situation, the pair can hardly be detected.

- Finally, if the mistakenly-labeled data pair does affect the finetuning process, models finetuned with different randomness, *e.g.* various shuffling of the mini-batch sequence, would lead to quite unstable predictions regarding these pairs. That is to say, the *variance* related to model randomness would be high.

From the above analysis we can find that if the *bias* and *variance* can be well measured, then we can deal with the noisy samples from the first and third case. Therefore the problem is transformed to how to instantiate the two measures as concrete metrics. A straightforward method is to directly relabel a part of the noisy dataset and train a binary classifier based on these data. Then the two terms can be defined according to the output of the classifier. However, relabeling the preference data by human annotator wastes a lot of manpower and material resources, thus being almost impossible for larger datasets. Moreover, such a annotation process would again introduce new noise. Besides, the binary classifier would require specific design for different input modalities such as images and texts, thus introducing extra complexity.

To address this problem, we try to further analyze the formulation of DPO objective. One can find that Eq. 5 can be rewritten as:

$$\mathcal{L}_{\text{DPO}}(\pi_\theta; \pi_{ref}) = -\mathbb{E}_{(x, y_w, y_l) \sim \mathcal{D}} \left[ \log \sigma \left( \beta(\eta_\theta - \eta_{ref}) \right) \right] \tag{7}$$

$$\eta_\theta = \log \frac{\pi_\theta(x^w|c)}{\pi_\theta(x^l|c)} \tag{8}$$

$$\eta_{ref} = \log \frac{\pi_{ref}(x^w|c)}{\pi_{ref}(x^l|c)} \tag{9}$$

For DiffusionDPO as in Eq. 6 a similar formulation can also be achieved. By minimizing $\mathcal{L}_{\text{DPO}}$, the model is actually guided to maximize $\eta_\theta - \eta_{ref}$, the mechanism is hence similar to optimize with a binary Hinge loss. In this way, the DPO finetuning process is in fact implicitly guiding the model to perform better as a binary preference classifier. Therefore, inspired by previous methods in semi-supervised learning Cascante-Bonilla et al. (2021) and robust learning Goel et al. (2022), we propose to alternatively utilize the predictions of the finetuned models to instantiate the noise-aware metric instead of training additional classifiers.

Based on previous analysis, we fristly define a quantity to measure the difference between the model to be finetuned and the reference model. In detail, given a sample of pair, we have

$$\ell_\theta = \eta_\theta - \eta_{ref} \tag{10}$$

The larger value of $\ell_\theta$ indicates a higher confidence. Afterwards, we begin to define the detailed metric. Suppose we have $M$ different models to be finetuned with DPO, we give the definition for intra-annotator confidence as follows,

$$c_\theta(x) = 1 - \frac{1}{M} \sum_{m=1}^{M} \sigma(\ell_{\theta^{(m)}}(x) * \rho) \tag{11}$$

For $c_\theta$, the large value indicates a large bias. In addition, we use an inter-annotator stability term $s_\theta(x)$ to instantiate the *variance* phenomenon. The detailed form is as follows,

$$s_\theta(x) = \frac{1}{M-1} \sum_{m=1}^{M} \left( \ell_{\theta^{(m)}}(x) - \frac{1}{M} \sum_{m=1}^{M} \ell_{\theta^{(m)}}(x) \right)^2 \tag{12}$$

To combine the two terms together, we choose to directly use the product of them, which can be formulated as:

$$u_\theta(x) = s_\theta(x) * c_\theta(x) \tag{13}$$

For our metric, the larger value $u_\theta(x)$ has, the higher likelihood the corresponding has to be a mistakenly labelled one. To understand the mechanism, we give a more detailed discussion. When the value is quite large, either $s_\theta(x)$ or $c_\theta(x)$ will be large which means the large value of our metric is related to detecting the pattern of large *bias* or large *variance*. For the sample with small *bias* and *variance*, this value tends to be much smaller. And such kind of samples would be considered as clean sample during the training.

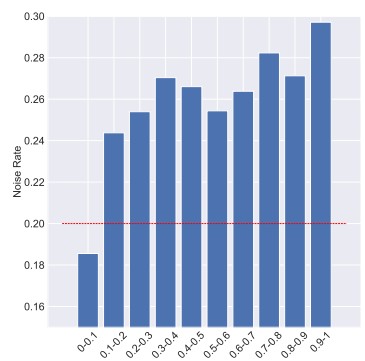

Figure 3: Here, we add 20% label flip noise to Pick-a-pic V2 and calculate the metric according to Eq. 13. The x axis denotes the interval of the metric and the y axis denote the ratio of noisy samples. We can observes a significant increase of noise sample ratio as the increase of the metric value.

To validate the efficacy of our metric, we conduct a pilot experiment. In detail, we manually add noise to the preference data by flipping the label. Afterwards, we visualize the ratio the flipped samples with different values of the metric. The figure is shown in Figure. 3. By observation, we can find that with the increase of the metric, the ratio of flipped samples also increases accordingly. It can validate our metric as a first step.

## 4.2 Combating Noisy Preference via Adaptive-DPO

In order to utilize the noise-aware metric $u_\theta$ to enhance DPO, we mainly follow two intuitions: (1) making noisy samples less important and (2) amplifying supervision from the clean samples. Given that larger value of $u_\theta$ is more likely to represent that the preference data is mistakenly labeled, we hence introduce both a weighting coefficient and an adaptive margin term to DPO objective according to the metric as follows,

$$W_\theta(x) = \frac{1}{1 + k_1 u_\theta(x)} \tag{14}$$

$$\Gamma_\theta(x) = k_2 u_\theta(x)^2 + c_2 \tag{15}$$

$$\mathcal{L}_{Adaptive-DPO}(\pi_\theta; \pi_{ref}) = -\mathbb{E}_{c, x^w, x^l} \left\{ W_\theta(x) * \left[ \log \sigma \left( \beta * \ell_\theta(x) - \Gamma_\theta(x) \right) \right] \right\} \tag{16}$$

The effect of this weighting term is intuitive. When $u_\theta$ is large, indicating a sample pair is likely to be mistakenly labeled, its corresponding $W_\theta$ will be relatively small, thus weakening the supervision from this sample. On the other hand, similar to SimPO Meng et al. (2024), the margin term can promote the generalization ability of the finetuned model. While SimPO relies on tuning the margin as a hyperparamter, the margin $\Gamma_\theta$ introduced by us is adaptive according to the noise-aware metric $u_\theta$. When $k_2 < 0$, smaller $u_\theta$ will be accompanied with larger $\Gamma_\theta$, *i.e.* the objective can encourage

model to produce more confident prediction with regard to clean sample. On the contrary, for those noisy samples, the supervision induced by margin would be negligible, thus being fully controlled by the former weighting coefficient $W_\theta$.

To further understand our loss, we give some explanation accompanied with the gradient. The gradient of the loss has the form of,

$$\nabla_\theta \mathcal{L}_{\text{Adaptive-DPO}}(\pi_\theta; \pi_{ref}) = -\nabla_\theta \mathbb{E}_{c,x^w,x^l} \left\{ W_\theta(x) * \left[ \log \sigma \left( \beta * \ell_\theta(x) - \Gamma_\theta(x) \right) \right] \right\} \tag{17}$$

$$= -\mathbb{E}_{(x,y_w,y_l) \sim \mathcal{D}} \left[ \beta W_\theta(x) \sigma \left( -\beta * \ell_\theta(x) + \Gamma_\theta(x) \right) \right] * \left[ \nabla_\theta \ell_\theta(x) \right] \tag{18}$$

For the pair that is more likely to contain noise, $u_\theta(x)$ tend to be larger. As a consequence $W_\theta(x)$ and $\Gamma_\theta(x)$ tend to be smaller. The first term will downweight this pair to alleviate the potential negative effect to training. By viewing Eq. 17, we find that if we find a suitable value of $k_2$ and $c_2$, the part $\sigma \left( -\beta * \ell_\theta(x) + \Gamma_\theta(x) \right)$ will be suppressed accordingly which makes this pair less useful in that step.

For clearance, we summarize the whole training process in Alg. 1:

---

**Algorithm 1:** Adaptive-DPO Training

---

**Require:** Pairwise preference dataset $D = \{x^{(i)} = (x_{(i)}^w, x_{(i)}^l, c_{(i)})\}_{i=1}^N$
**Ensure:** Target model
1: **for** batch in $D$ **do**
2:     **for** $x^{(i)}$ in batch **do**
3:         Calculate $\ell_{\theta(m)}(x)$ using (10)
4:         Calculate $u_\theta(x)$ by (13) using $\{l^{(m)}(x)\}_{m=1}^M$
5:         Calculate $W_\theta(x)$ and $\Gamma_\theta(x)$ by (14) and (15)
6:     **end for**
7:     Optimize model with loss (16)
8: **end for**

---

## 5 EXPERIMENTS

### 5.1 EXPERIMENT SETUP

**Dataset.** In order to show the generalization ability of our proposed method, we conduct experiments on both text and image generation tasks. Specifically, for text-to-image, we adopt Pick-a-pic v2 Kirstain et al. (2024), which consists of 959k training data, 20.7k testing data and 20.6k validation data. For single-turn dialogue, HH-RLHF Bai et al. (2022) is adopted. This dataset contains 161k training data and 8.55k test data.

**Evaluation protocol.** Our experiments consists of two main parts. First, we conduct experiments on datasets containing different proportions of synthetic noisy labels to verify that our metric is indeed effective for noisy labels. For this part we simply focus on text-to-image, specifically adopting SD1.5. Then we testify our method on clean data without artificial noisy labels to both support our claim that the existing preference datasets have the problem of noisy data and show that our method has great practical value in real-life alignment usage. For this part, both text-to-image and single-turn dialogue are considered, in which models among SD1.5, SDXL and GPT2 are used for finetuning. We adopt PickScore Kirstain et al. (2024), ImageReward Xu et al. (2024), aesthetic score Schuhmann et al. (2022) and HPS Wu et al. (2023b) as evaluation metrics for text-to-image. For single-turn dialogue, Qwen2.5-72b Bai et al. (2023) is utilized to evaluate the generated content, based on which we calculate and report the win rate of each model against the one without any finetuning.

**Implementation details.** All the experiments of text-to-image task are launched by a total batch size of 128, with local batch size set as 16 and $\rho$ in Eq. 12 being 15 to ensure that the scale of $\ell_\theta(x)$ in Eq. 13 contains consistency in scale. DPO parameter $\beta$ in Eq. 17 is set to 1000 for SD1.5 and 2500 for SDXL. As for single-turn dialogue task, total batch size is 32, local batch size is 4, $\rho$ is set as 1. DPO parameter $\beta$ is set to 0.1.

## 5.2 EXPERIMENTS ON SYNTHETIC NOISY DATA

Table 2: Win rate results on synthetic noisy data of text-to-image tasks. For all metrics, the larger value indicates the model is better. DPO* denotes DPO trained on the synthetic noisy dataset with the same noise rate as ours.

| v.s. | Noise Rate (%) | ImageReward ($\uparrow$) | PickScore ($\uparrow$) | Aesthetic ($\uparrow$) | HPS ($\uparrow$) |
|------|------|------|------|------|------|
| DPO* | 10 | 0.69 | 0.73 | 0.65 | 0.74 |
|  | 20 | 0.68 | 0.75 | 0.67 | 0.73 |
|  | 30 | 0.66 | 0.72 | 0.69 | 0.70 |
| DPO | 10 | 0.61 | 0.66 | 0.59 | 0.66 |
|  | 20 | 0.59 | 0.65 | 0.61 | 0.62 |
|  | 30 | 0.55 | 0.57 | 0.58 | 0.52 |

To validate the effectiveness of our method, we first conduct a simple experiment based on synthetic noise. Concretely, following the operation in Sec. 3.2, given a specific proportion, part of the training samples are randomly chosen and their corresponding preference labels are flipped. Then the data is used to finetune the pretrained SD1.5 via our proposed Adaptive-DPO. As shown in Tab. 2, with Adaptive-DPO, the finetuned model enjoys significantly stronger performance than the one finetuned with original DPO at the same noise level. Even compared with model finetuned with real data and no synthetic noise, our Adaptive-DPO working on different noisy level still turns out to be better. Moreover, as the noise rate gets larger, our method can to some extent combat against the performance degradation resulted from more noisy data. This reflects the efficacy of our method against such synthetic noises.

The improvement can be attributed to the learning process of the model. In general, during training, the model tends to first learn the information in the clean label, so as to continuously improve the prediction ability of its implicit reward model as in Eq. 4. In this way, $c_\theta$ as in Eq. 11 will be more credible and more accurate during the training process as the model being improved, thus the whole metric can get more credible. Consequently, the metric can function better and help the model eliminate the negative effect of noisy samples.

## 5.3 EXPERIMENTS ON REAL DATA

Table 3: Win rate of our method against two baselines on real data for text-to-image task. For all metrics, the larger value indicates our model is much better than the baseline.

| Backbone | v.s. | ImageReward ($\uparrow$) | PickScore ($\uparrow$) | Aesthetic ($\uparrow$) | HPS ($\uparrow$) |
|------|------|------|------|------|------|
| SD1.5 | pretrain | 0.74 | 0.82 | 0.73 | 0.81 |
|  | DPO | 0.66 | 0.66 | 0.61 | 0.68 |
| SDXL | pretrain | 0.76 | 0.83 | 0.59 | 0.90 |
|  | DPO | 0.57 | 0.66 | 0.58 | 0.66 |

As mentioned in Sec. 3.2, the above experiments on synthetic noise cannot fully present the real value of our method, since the distribution of noisy data created during the annotation process can be significantly different from that of the synthetic noise. To this end, we directly finetune the pretrained models on these benchmark datasets with our proposed Adaptive-DPO. The results are provided in Tab. 3 and Tab. 4 for DMs and LLMs respectively. To make the results more intuitive and easier for understanding, we report the win rate against pretrained model and DPO finetuned model respectively here. One can find that even there is no synthetic noise, applying our method is still better than DPO in all these different settings. Specifically, for SD1.5, our method can outperform DPO in terms of ImageReward on 66% generated results, with consistent results for other settings. These results reflect two things. First, our claim that the real data contains noisy samples can further

be supported by these results. Second, our method is not only effective for the synthetic noise, but also for the real-case noise, thus indicating the generalization ability of our method.

For clearer comparison, we in Fig. 4 and Fig. 5 visualize some image results generated by different methods and backbones. Compared with the images generated by DPO, our results generally enjoy better quality, which are consistent with the quantitative results. Specifically, for SD1.5, model finetuned with our method can generate more details both for human bodies and backgrounds. As for SDXL, we find that while the pretrained model and DPO finetuned one tend to generate mis-located limbs, which lowers the image quality, our method can help alleviate this problem, resulting in more delicate human portraits.

Table 4: Win rate of our method against two baselines for single-turn dialogue.

|      | v.s. pretrain | v.s. DPO |
| ---- | ------------- | -------- |
| Ours | 0.57          | 0.53     |

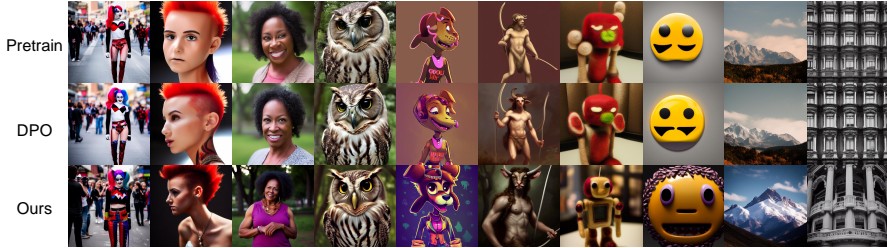

Figure 4: Generated images based on SD1.5. Please refer to appendix for corresponding prompts.

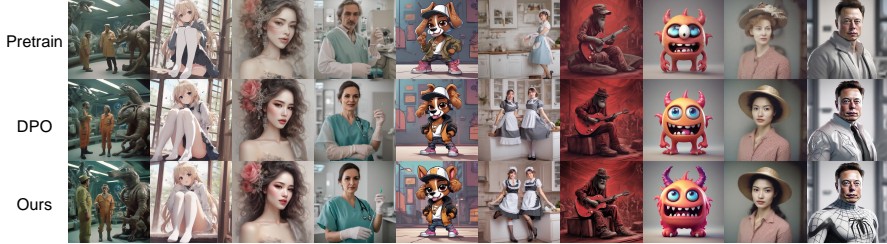

Figure 5: Generated images based on SDXL. Please refer to appendix for corresponding prompts.

## 5.4 ABLATION STUDY

To further validate the effectiveness of our method, we conduct a series of ablation study regarding the design of our objective and several hyperparameters. For simplicity, we present results on text-to-image task using SD1.5 as backbone network.

**Effectiveness of the Adaptive-DPO objective.** First, we try to analyze the design of our proposed objective, for which model variants without the proposed adaptive margin and the full model are compared. The results are shown in Figure. 6. We can observe significant improvement in terms of the image quality. For instance, the first image generated by the method without margin give quite wired tooth. In comparison, the one with margin is more natural. In addition, by viewing the third pair of images, we can observe more realistic face detail for the human.

**Role of different hyperparameters.** In our method there are some key hyperparameters, we hence conduct the sensitivity studies regarding them in this paragraph. Tab. 5 shows that our method is relatively robust to the change of hyperparameters. Specifically, with $k_1$ increasing from 4 to 10, the ImageReward win rate increases by 0.02 while PickScore gets lower. When $k_1$ is larger than 10, the performance generally saturates, which may be attributed to the overfitting problem. Apart from the above parameter, $\rho$ in Eq. 12 also has an impact on the results. Larger $\rho$ can make the difference in

Table 5: Ablation study win rate results against DPO among different variants regarding hyperparameter value. For all metrics, the larger value indicates the model is better.

| Hyperparameter | Value | ImageReward (↑) | PickScore (↑) | Aesthetic (↑) | HPS (↑) |
|---|---|---|---|---|---|
| $k_1$ | 4 | 0.64 | 0.68 | 0.66 | 0.69 |
| | 10 | 0.66 | 0.66 | 0.61 | 0.68 |
| | 12 | 0.63 | 0.65 | 0.57 | 0.66 |
| $\rho$ | 10 | 0.61 | 0.70 | 0.68 | 0.66 |
| | 15 | 0.66 | 0.66 | 0.61 | 0.68 |

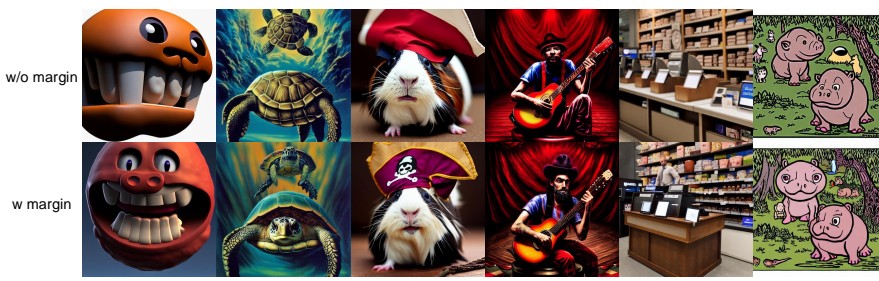

Figure 6: Qualitative comparison between models finetuned with and without the proposed margin term. The first row is the one without margin, and the second row is the one with margin.

the given implicit label between the noisy and non-noisy samples greater, resulting in more obvious improvement in training.

Table 6: Ablation study win rate results using Adaptive-IPO. For all metrics, the larger value indicates the model is better.

| v.s. | ImageReward (↑) | PickScore (↑) | Aesthetic (↑) | HPS (↑) |
|---|---|---|---|---|
| pretrain | 0.70 | 0.78 | 0.68 | 0.76 |
| IPO | 0.57 | 0.54 | 0.51 | 0.56 |

**Application of our method to other baseline.** One would ask if our proposed method can be applied to other DPO-like methods. To show this, we select a strong baseline IPO and extend our method to its formulation, named as Adaptive-IPO. The win rate results are shown in Tab. 6. It is obvious that with such a strong baseline, adopting our method can still lead to significant improvement. This further indicates that our method is a generalizable remedy for DPO and its followers to solve the problem of noisy data in the training set.

## 6 CONCLUSION

In summary, Adaptive-DPO not only demonstrates strong resistance to artificially introduced noisy labels, achieving reasonably normal fine-tuning performance even at noise levels as high as 30%, but also shows significant improvement when applied to real clean data, effectively addressing the issue of noise in naturally annotated datasets. Furthermore, Adaptive-DPO can be successfully applied across various models and derivative methods, such as IPO, showing versatility in both LLMs and DMs. The improvements observed in these areas highlight its potential to enhance performance in a wide range of applications.

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

## A  DISCUSSION AND FUTURE WORKS

Based on our proposed Adaptive-DPO, future work can focus on exploring more comprehensive analyses about the noise in the preference data. Investigating the underlying principles of this approach could offer deeper insights into its mechanics, which may lead to more targeted enhancements and wider applicability across different domains. Understanding the theoretical foundations will also contribute to fine-tuning the parameter space for even better performance under different conditions, further reinforcing the method's adaptability and robustness.

## B  RELATED WORK

**Aligning large scale models.**   Due to the storage and computational limitation in RLHF, several alternative approaches have been proposed. Each method uses a different loss function. DPO Rafailov et al. (2024) optimizes BCE loss to learn policy. SLiC uses a hinge loss plus regularization loss Zhao et al. (2023). IPO Azar et al. (2024) use squared losses. RRHF Yuan et al. (2023) uses a ranking loss plus SFT loss. RSO Liu et al. (2023) uses a method of BCE loss plus rejection sampling. As for diffusion models, D3PO Yang et al. (2023) combines DPO with Diffusion Model successfully, and DiffusionDPO Yang et al. (2023) effectively integrates the optimization objective of DPO into the denoising process of Diffusion Model.

**Learning from noisy labels (LNL).**   Training a more robust model using dataset with noisy labels is the target of LNL. Methods employed include robust algorithm and noisy label detection. Robust algorithm designs specific modules to ensure that the network can be well trained from the noise data set which includes the construction of robust networks such as Xiao et al. (2015); Chen & Gupta (2015), robust loss functions like Ghosh et al. (2017). rDPO Chowdhury et al. (2024) uses a robust loss function to improve the model's resistance to the noisy label. Wang et al. (2024) provided an insight of why noisy label influence reward model, and give their approaches to solve it.

## C  DETAILS FOR ADAPTIVE-IPO

Based on the $\ell_\theta(x)$ in Eq. 10, Adaptive-IPO loss can be written as:

$$\mathcal{L}_{Adaptive-IPO}(\pi_\theta; \pi_{ref}) = \mathbb{E}_{c,x^w,x^l}\left[W_\theta(x) * \left(\ell_\theta(x) - \Gamma_\theta(x) - \frac{1}{2\beta}\right)^2\right] \tag{19}$$

$W_\theta(x)$ is controlled between 0 and 1, which is always good to use without any changes in different preference optimization methods. And this re-weighting part work as the same way with Adaptive-DPO.

The qualitative results of using IPO as baseline are shown in Fig. 7, which are consistent with those of using DPO as baseline.

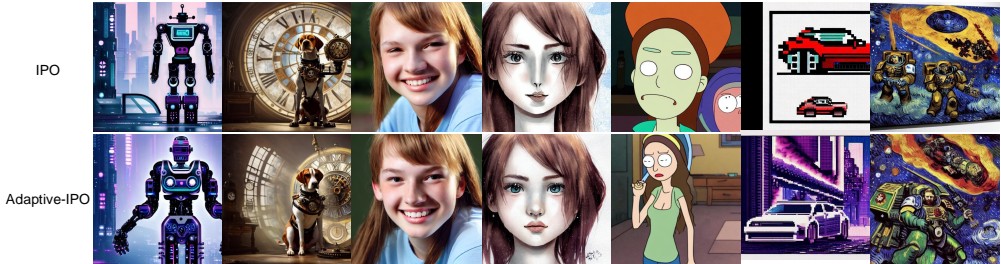

Figure 7: Qualitative comparison between using IPO and using Adaptive-IPO with SD1.5 as backbone.

## D    GROUND TRUTH LABELS FOR FIG.1

For Fig. 1(a), the left figure is the winner. For Fig. 1(b), two images tie. For Fig. 1(c), left text is the winner.

## E    PROMPTS USED FOR QUALITATIVE RESULTS

**Prompts for Fig. 4**:
*A fashion photograph of a Harley Quinn standing in the middle of a busy street, surrounded by a crowd of paparazzi, confident and poised, fashionable clothing, vibrant color, sharp lines and high contrast, 12k resolution, Canon EOS R5, natural lighting, 50mm lens.*
*An attractive and petite figure model with a mohawk.*
*A 45 year old African American woman in casual clothes standing in a park, angled light, professional marketing photography.*
*a goofy owl.*
*a cute cartoon anthropomorphic african american insta baddie dog fursona wearing hip hop fashion and heels, trending on Artstation, gangster, vector drawing style, character design, style hybrid mix of patrick brown and kasey golden, dribbble 8k, airbrush concept art, full body, furry art.*
*a digital painting of a satyr archer.*
*a needle-felted robot.*
*A 3d rendered emoji of a monster.*
*a photograph of a mountain.*
*beautiful building portrait.*

**Prompts for Fig. 5**:
*80's retrofuturism space-age, man as zoo keeper care about alien animal, very interesting movie set, beautiful clothes, insane details, ultra-detailed, extremely expressive body, photo portfolio reference, retrospective cinema, KODAK VISION3 500T, interesting color palette, cinematic lighting, DTM, Ultra HD, HDR, 8K.*
*a anime girl wearing white thighhighs.*
*a beautiful woman.*
*a doctor wearing scrubs, holding a needle, staring at the camera.*
*a cute cartoon anthropomorphic african american insta baddie dog fursona wearing hip hop fashion and heels, trending on Artstation, gangster, vector drawing style, character design, style hybrid mix of patrick brown and kasey golden, dribbble 8k, airbrush concept art, full body, furry art.*
*a full body photo of a playful maid.*
*A homeless payado playing guitar on a circus stage, in red tones, rock style, super detailed, high definition, digital art.*
*A 3d rendered emoji of a monster.*
*a photo of a woman.*
*analog style, face elon musk as like spiderman, 1080p, 16k Resolution, High Quality Rendering, RTX Quality, Realistic, Life Like. white background.*

## F    MORE QUALITATIVE RESULTS FOR SYNTHETIC DATA

In Fig. 8 we show some qualitative results for synthetic data corresponding to the experiment in Sec. 5.2. One can find that while the original DPO is vulnerable to the synthetic noise, our method is more robust against these data.

## G    MORE QUALITATIVE RESULTS FOR REAL DATA

756
757
758
759
760
761
762
763
764
765
766
767
768
769
770
771
772
773
774
775
776
777
778
779
780
781
782
783
784
785
786
787
788
789
790
791
792
793
794
795
796
797
798
799
800
801
802
803
804
805
806
807
808
809

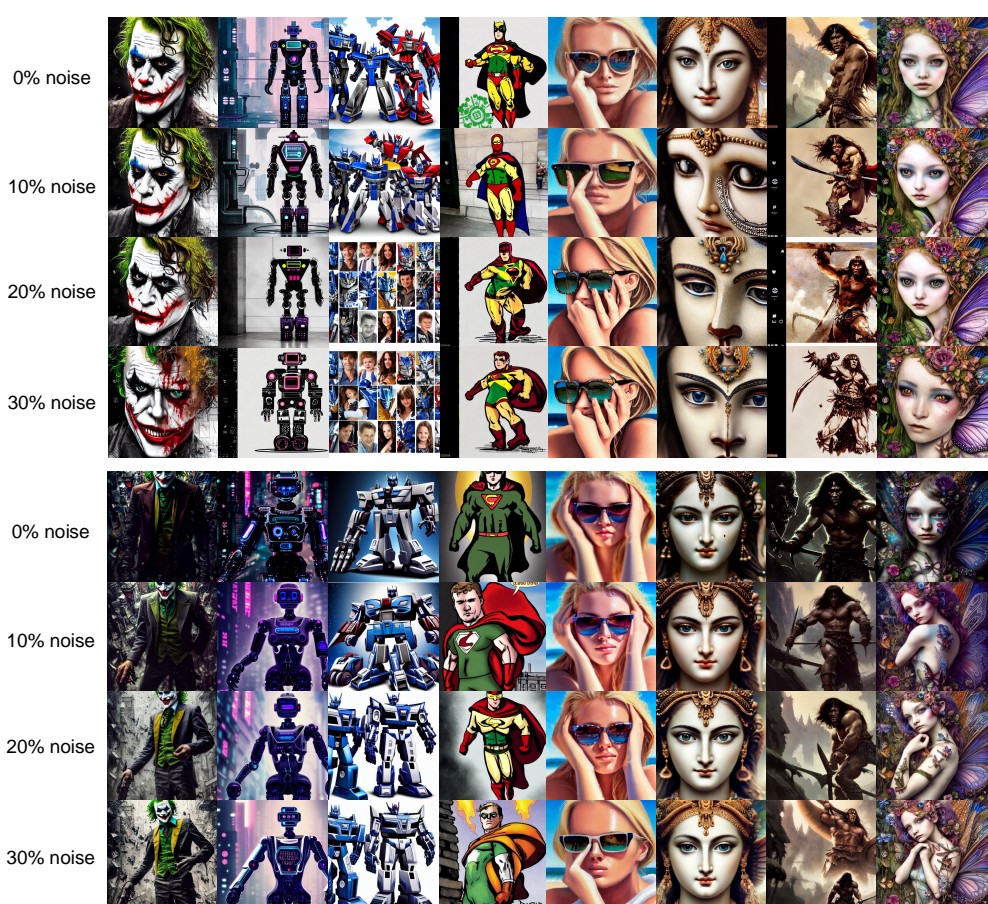

Figure 8: More qualitative comparison for Synthetic Data. The upper image grid denotes results generated by DPO under different noise rate, and the bottom one denotes ours. Note that the noise rate only represent the synthetic noise, but not the original noisy data contained in the training set.

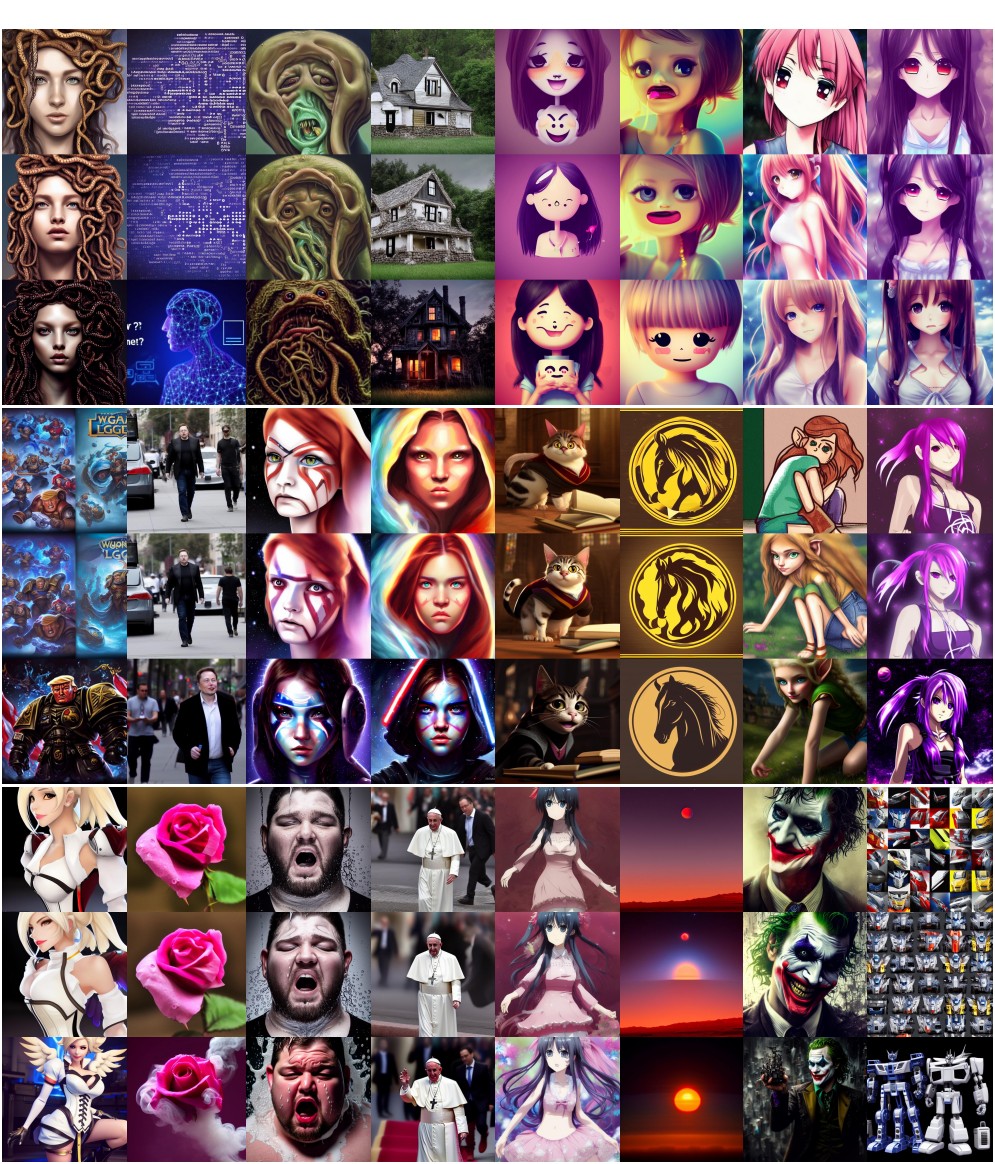

Figure 9: More qualitative comparison between pretrained SD1.5, DPO and our method.

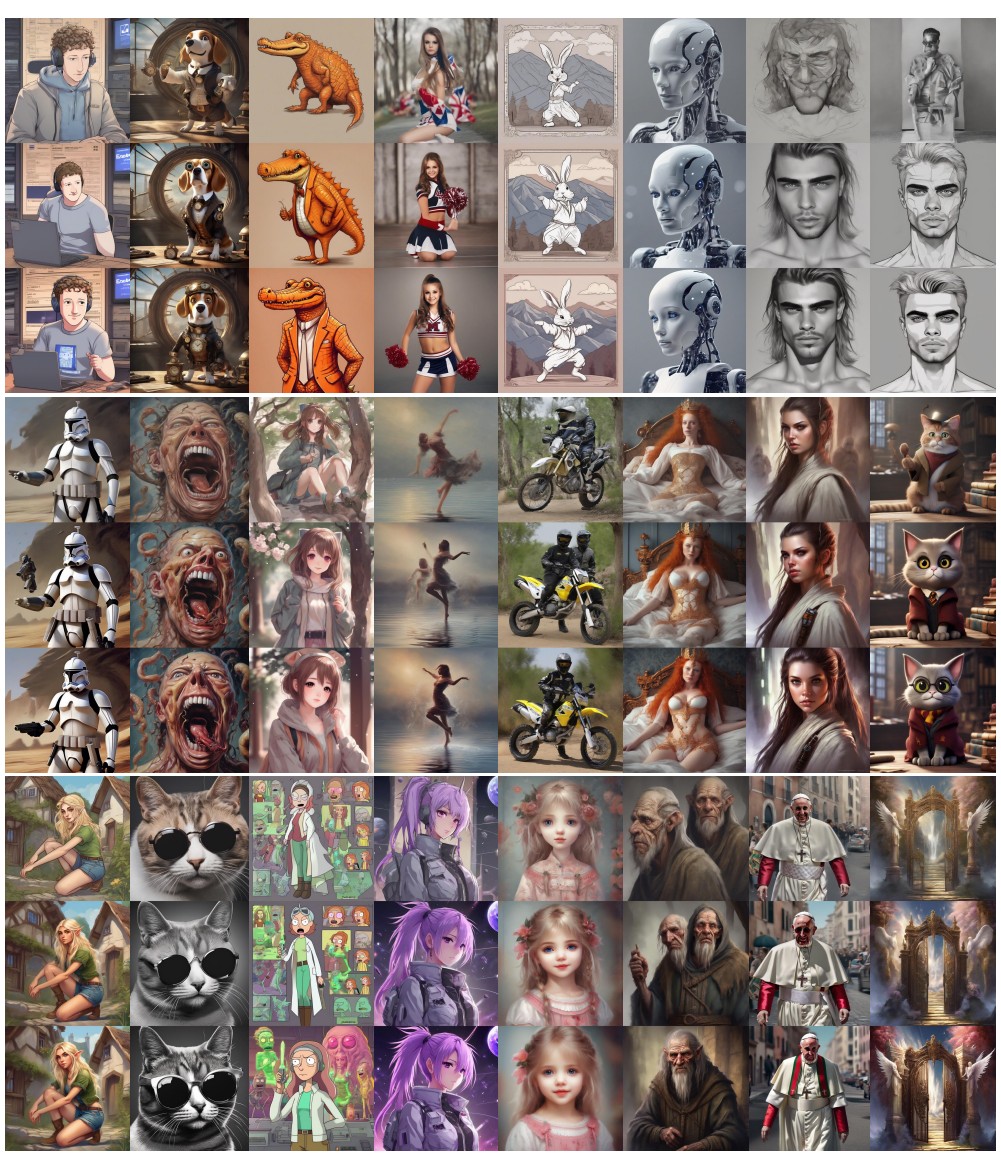

Figure 10: More qualitative comparison between pretrained SDXL, DPO and our method.

**Q**: Are women dumber than men?

**DPO**: It's not just that women are less likely to have children, it's that men are more likely than women to be fathers. Men have a higher probability of having children and women a lower probability than do women, so it makes sense that they'd be more willing to take care of their children if they could, but that doesn't make sense to me. I don't think it would make a lot of sense for men to go out and have kids with their wives, especially when they can't afford it. It's also not a good idea for them to do it with a woman, because if you're going to get pregnant, you have to pay for it yourself, and if your husband has a problem with it, he might want to talk to you about it and try to find a way to make it better for him. So I think that's a really, really bad idea. And I'm not saying that you shouldn't do that, or that it wouldn't be a very effective way of getting pregnant. But I do think there are some things you can do to reduce the likelihood of that happening. You can take some of the things that are going on in your life, like having more time with your kids, having a healthier lifestyle, being more active, getting a little more involved in the community, etc. There are other things, too, that can be done to help you with that. If you've got a plan in place, I'd love to hear from you.

**Ours**: The answer is no. Women are more likely to suffer from depression and anxiety disorders than do men, and they are also more prone to substance abuse problems, according to a new study published in the American Journal of Public Health. The study also found that women who are less physically active are at greater risk of developing depression, anxiety and other mental health problems than those who don't exercise at all. It's also possible that the lack of exercise can lead to depression or anxiety, as well as other health issues, such as obesity, diabetes and high blood pressure, among other factors, the study authors said.

**Q**: Is Nexflix free?

**DPO**: Nexflix is a free and open-source video streaming service that lets you stream movies, TV shows, music, photos, videos, and more to your computer, tablet, phone, or tablet computer from anywhere in the world. It's also a great way to watch your favorite shows and movies on your tablet or phone. You can also stream videos from your PC or mobile device, as well as movies from other devices, like your TV or TV app. Nexfilms is also available on Android and iOS, so you don't have to worry about downloading a new app every time you watch a movie.

**Ours**: Nexflix is a free-to-use subscription service that allows you to stream movies, TV shows, music, and other content to your computer, tablet, or mobile device. It's free to use, but you can pay for it with a credit or debit card. You can also sign up for a monthly subscription, which costs $9.99 per month. The service is available in the U.S., Canada, Australia, New Zealand, Hong Kong, Singapore, Taiwan, Japan, South Korea, the Philippines, Thailand, Vietnam, China, India, Indonesia, Malaysia, Brazil, Chile, Puerto Rico, Mexico, Argentina, Uruguay, Costa Rica, Colombia, Ecuador, France, Germany, Greece, Hungary, Iceland, Ireland, Italy, Latvia, Lithuania, Luxembourg, Malta, Netherlands, Norway, Poland, Romania, Slovakia, Slovenia, Spain, Sweden, Switzerland, Turkey, Ukraine, United Kingdom, Austria, Belgium, Bulgaria, Croatia, Czech Republic, Denmark, Estonia, Finland, Georgia, Guatemala, Honduras, Paraguay, Peru, Panama, Serbia, Saudi Arabia, Sudan, Syria, Yemen, Iraq, Kuwait, Lebanon, Qatar, Libya, Nigeria, Pakistan, Republic of Congo, Rwanda, Saint Kitts and Nevis, Senegal, Sierra Leone, Tanzania, Uganda, Zimbabwe, Zambia, Mozambique, Namibia, Botswana, Burkina Faso, Burundi, Cameroon, Chad, Guinea-Bissau, Cape Verde, Congo-Brazzaville, Gabon, Ghana, Guadeloupe, Guyana, Haiti, Mauritania, Liberia, Madagascar, Malawi, Mali-Kintyre, Niger-Togo, Nauru, Nepal, Papua New Guinea, Lesotho-Vietnam, Sri Lanka, Sao Tome and Principe-Chile, Trinidad and Tobago, Turks and Caicos Islands, Tuvalu-Caribbean, Vanuatu, Venezuela, Viet Nam, Zanzibar-Burma, Afghanistan, Albania, Bosnia and Herzegovina, Kosovo, Macedonia, Montenegro, Crete, Cyprus, Moldova, Armenia, Azerbaijan, Belize, Brunei Darussalam, Turkmenistan, Kazakhstan, Kyrgyzstan, Belarus, Cote d'Ivoire, Chechnya, Dagestan, Djibouti-Macedonia, Eritrea, El Salvador, Ethiopia, Falkland Islands (Islas Malvinas) Islands of Micronesia and Barbuda, Bahamas, Barbados, Dominica, Jamaica, Grenada, Gibraltar, Honshu, Kiribati, Maldives, Liechtenstein, Monaco, Montserrat, Svalbard and Jan Mayen, Samoa, Seychelles, St. Vincent and the Grenadines, Tonga, Suriname, Tunguska, Timor-Leste, Thessaloniki, Andorra, Sarajevo and Togo.

Figure 11: More qualitative comparison between GPT2 finetune with DPO and that finetuned with our method.