# OpenReview forum: "Combating inherent noise for direct preference optimization"
_ICLR.cc/2025/Conference — Submitted to ICLR 2025_

### Official Review · Reviewer_ouSA · 2024-10-30

**Soundness:** 3
**Presentation:** 2
**Contribution:** 2
**Rating:** 5
**Confidence:** 4

**Summary:**

This paper focuses on addressing the issue of data noise in DPO alignment. It proposes a metric to evaluate the level of noise and, based on this metric, introduces an Adaptive-DPO loss function to reduce the impact of noisy samples and amplify the influence of clean samples.

**Strengths:**

1. The idea shows good innovation and the formulas are comprehensive.
2. The proposed new loss function demonstrates significant advantages over the original DPO on real datasets.

**Weaknesses:**

1. For Table2, I understand that the win rate of Adaptive-DPO vs. DPO* should increase as the noise rate increases, assuming Adaptive-DPO can resist the impact of noise better than DPO can. Could you explain why the indicators here do not show this trend?

2. Regarding "as the noise rate gets larger, our method can to some extent combat against the performance degradation resulted from more noisy data," I think you need an experiment comparing a model trained with DPO on synthetic data at different levels of noise rates with a model trained with DPO on real data to better support this point.

3. For the ablation study, I think it's missing the separate ablation results for W_{\theta}(x) and \Gamma_{\theta}(x), for both language models and text-to-image models, rather than just case study comparisons like in Figure 6.

4. Format Error: In Table 1, there is a wrong bold section of the PickScore column.

**Questions:**

N/A

---

### Official Review · Reviewer_eT9w · 2024-11-03

**Soundness:** 2
**Presentation:** 2
**Contribution:** 2
**Rating:** 3
**Confidence:** 4

**Summary:**

This paper proposes two noise-aware metrics (i.e., intra-annotator confidence and inter-annotator stability) to improve the performance of DPO under preference noise. Then, the authors introduce the proposed metrics to the vanilla DPO loss to reduce the impact of noisy samples. Experiments on the tasks of text-to-image generation and single-turn dialogue show the effectiveness of the proposed method to some extent.

**Strengths:**

1. The problem of aligning large models with noisy feedback is important.
2. The idea of computing the bias and variance of confidences from multiple annotators is interesting.

**Weaknesses:**

My concerns are as follows.
1. The claims that “the quality of preference data used in DPO training has been largely overlooked” (Lines 14-15) and “an often-overlooked aspect of the training process is actually the preference data used for training” (Lines 36-37) are incorrect. In fact, many existing studies [1,2,3] have noticed the prevalence of noise in the preference alignment and have proposed methods to address the challenges brought by noisy preferences.
2. Many important baselines for LLM alignment with noisy preferences are missing, such as C-DPO [2], R-DPO [3]. This point, together with the previous one, significantly hurts the professionalism and reliability of this work.
3. The motivation of the proposed Adaptive-DPO is based on a assumption---the model being trained tends to first learn the information in the clean label (as stated in Lines 404-406). However, this hypothesis requires solid theoretical or empirical support. PLEASE NOTE THAT I acknowledge that this phenomenon is common in some traditional classification tasks, but in the area of preference learning, this may not be true (i.e., DPO loss).
4. Theoretical or empirical analysis are needed to demonstrate the reliability of the proposed metric $u_\theta(x)$. As shown in Figure 3, when $0.1 \leq u_\theta(x) \leq 1$, the noise rate stably stays in the range of $(0.24, 0.30)$, which means that more than 70% samples are clean. Therefore, the metric seems to be ineffective in identifying noise.
5. AdaptiveDPO requires simultaneously training $M$ different models to compute the noise-aware metrics, i.e., bias and variance, which is expensive in real-world applications. Besides, the value of $M$ in experiments are not mentioned in Section 5.1.
6. The authors may want to follow existing studies [4,5,6,7] to use GPT-4 as the evaluator, which has already become a standard experimental setting in the preference alignment.
7. The authors may want to report the numbers of scalar metrics of model performance, such as ImageReward, PickScore, Aesthetic, and HPS, rather than just providing win rates under these metrics, as shown in Tables 2, 3, 5-6.
8. The authors may want to explain why the win rate of Adaptive-DPO vs DPO* (in Table 2) decreases as noise rate increases. This may indicate that Adaptive-DPO is ineffective under noisy preferences.
9. Many notations are used without explanation, such as the $\rho$ in Eq. (11), the $k_1, k_2, c_2$ in Eq. (15). Besides, what is the relationship between $c$ and $c_2$? Why do the authors use $\ast$ to denote multiplication operation? Although this is the last point, the weaknesses in notations are not minor. This is because preference alignment (whether RL-based or RL-free) is a direction based on theoretical derivation, and rigorous and clear mathematical presentation is very necessary.

[1] Impact of Preference Noise on the Alignment Performance of Generative Language Models

[2] A Note on DPO with Noisy Preferences & Relationship to IPO

[3] Provably Robust DPO: Aligning Language Models with Noisy Feedback

[4] SimPO: Simple Preference Optimization with a Reference-Free Reward

[5] Zephyr: Direct Distillation of LM Alignment

[6] Length-Controlled AlpacaEval: A Simple Way to Debias Automatic Evaluators

[7] Judging LLM-as-a-Judge with MT-Bench and Chatbot Arena

**Questions:**

Please see Weaknesses.

---

### Official Review · Reviewer_rzzD · 2024-11-04

**Soundness:** 2
**Presentation:** 3
**Contribution:** 2
**Rating:** 3
**Confidence:** 3

**Summary:**

This paper discusses the problem of noisy labels in preference data used for DPO. The authors propose to incorporate a noise-aware metric into the DPO objective to identify and mitigate the impact of noisy data. The experiment results show that the proposed method effectively improves the performance in visual and textual generation tasks.

**Strengths:**

1. The starting point of the paper is interesting and important to the field. The problem of noisy labels in preference data is a novel field and needs to be explored.
2. The presentation of algorithm in Equation 11,12,13 and Algorithm 1 is well explained, reasonable, and easy to understand. Rich exploratory analysis and experiments help to understand the problem and the proposed solution.

**Weaknesses:**

1. The paper needs more careful consideration of the problem of noisy labels in preference data. The authors mentioned that the noisy labels could due to the limited generalization ability of the machine annotators, inaccuracies from careless or malicious annotators, and the lack of standardized judgment criteria (line 038-043).
- For the model annotator case, the authors do not provide enough explanations, citations, or analyses.
- The human annotator case is a tricky problem. There could be two types of annotator mistakes: one is the annotator is not careful enough, and the other is the annotator has an individual understanding of the problem. In the first case, we should have algorithms to detect and correct the mistake during the training. In the second case, we should train the model to cope with or be aware of such differences. The authors should discuss the two cases and place their algorithm in more detailed context.
- In the lack of standardized judgment criteria case, more recent work tends to carefully curate the preference data and provide a reliable standardization [Llama2, Beaver]. Citation of recent work could help to make the problem more relevant and up-to-date.

2. The scope of the paper is confusing. In the introduction and preliminary, the authors mention that the paper focuses on the text and image generation tasks. However, exploration experiments, main results, and ablation are only on the image generation task.

3. The presentation of the results are confusing. It is not mentioned why the score-based metrics in table 1,2,3,4,5,6 are presented as win rate rather than the raw scores.

4. The comparison with other methods mentioned in line 669 help to understand the significance of the proposed method.

5. The proposed method is a bit confusing. Please see the question part.

[Llama2] Touvron et al. Llama 2 Open Foundation and Fine-Tuned Chat Model, 2023.\
[Beaver] Dai et al. 2023 - Safe RLHF Safe Reinforcement Learning from Human, 2023.

**Questions:**

1. For the analysis of Figure, could you provide more details related to the background of the three annotators? Did you consider the individual bias of the annotators?

2. Could you provide concrete examples of the three situations explained in section 4.1? The second case is clear, but why would some examples be hard to provide practical supervision to the model and some can affect the fine-tuning process?

3. In line 271, the author mentioned "Suppose we have M different models to be finetuned with DPO,". What is the relationship between the number of models and the reliability of the metric in equation 11, 12? In the experiments, would this introduce extra comptuation cost?

4. In line 288, the author mentioned "The higher the value of the metric, the higher likelihood the corresponding has to be a mistakenly labelled one." Would it be the case that higher likelihood in eq 13, that is low confidence and high variance, means that the model lacks the knowledge and needs to be further finetuned? Regarding this question, is it reliable to use the model during the training to calculate the metric as in line 4 in Algorithm 1? Are the analysis in Figure 4 using the tuned model or the model during the training?


5. In line 323, the author mentioned "smaller u will be accompanied with larger gamma". Do you assume |c2|>>|k2| in this case? How are these two values selected in the experiments?

5. Table 2 shows the performance of the proposed method in different noise levels. Would the proposed methods hurt the performance in the 0 noise level?

**Details Of Ethics Concerns:**

The paper uses human annotator for the exploratory experiments. Details of this experiment needs to be provided.

---

### Official Review · Reviewer_tCWM · 2024-11-10

**Soundness:** 1
**Presentation:** 2
**Contribution:** 2
**Rating:** 3
**Confidence:** 5

**Summary:**

The paper addresses the issue of noisy labels in Direct Preference Optimization (DPO). To combat the preference noise, the work proposes a noise-aware metric that considers intra-annotator confidence and inter-annotator stability, and Adaptive-DPO loss function that enhances the standard DPO loss by reducing the influence of noisy samples and amplifying the impact of clean samples with the proposed noise-aware metric. Experimental results demonstrate the superiority of their proposed method in handling noisy preference data on both image and text generation tasks.

**Strengths:**

- The work deals with a practically important ML problem; handling data noise in preference learning.
- The work provides several interesting experiments, showing the possible negative effect of the preference label noise.
- The work covers two widely used generative models; LLMs and text-to-image diffusion models.

**Weaknesses:**

- The definition and problem setup are questionable; The reviewer is not convincing about the existence of a 'clean' preference, since the preference is mostly somewhat subjective. Even for the same data pair, the preference might differ across the users so we can not call it 'noisy' or 'noisy label'. I think a model can only learn the overall human preferences, and it should not be perfect in some ambiguous cases. In Sec 3.1, the authors argue that the inconsistency among different annotators is evidence of the noisy labels, but this definition is questionable. In my opinion, this ambiguous case should be clearly distinguished from the noisy preference data, which may be adversarially annotated for very easy cases.
- Similarly, in Sec 3.2, only synthetic noisy preferences are used to evaluate the vulnerability of DPO. Is this a frequently occurring scenario? and in my view, the performance drop in Table 1 seems not significant.
- In the methodology section, do the roles of the reweighting and adaptive margin modules overlap, and how do they synergize with each other? or are they complementary?
- Several expressions are vague; 1) In line 54, the author says "if users aim to avoid copyright-protected symbols to evade costly penalties, such deviations can create serious problems". What serious problems? more detailed examples should be explained. 2) Prompts used to generate images are missing in Figure 4,5,6; Since most of the reward scores, including ImageReward, PickScore, and HPS, not only measure the image quality but also assess the text-image alignment, the exact prompts need to present together.

**Questions:**

See weaknesses

---

### Meta-Review · Area_Chair_4t1b · 2024-12-19

**Metareview:**

This paper pointed out the noise label issue in preference data for DPO, and further proposed a noise-aware metric into the DPO objective, t reduce the influence of noisy samples and amplify the impact of clean samples. The paper received four negative ratings, and the authors did not respond to the reviews.

The major strengths is that the motivation and research question about noise label issue in DPO are interesting. The major weaknesses and reasons for reject are (1) the problem of noisy labels in preference data is not well defined, (2) some important DPO baselines are missing, (3) the scope is unclear as only text-to-image diffusion models are evaluated but both LLMs and text-to-image diffusion models are considered. The AC agreed with reviewers and recommend for reject.

**Additional Comments On Reviewer Discussion:**

The authors did not respond to reviewers' comments.

---

### Decision · Program_Chairs · 2025-01-22

Reject